# Common diseases alter the physiological age-related blood microRNA profile

Tobias Fehlmann [1], Benoit Lehallier [2], Nicholas Schaum[2], Oliver Hahn[2], Mustafa Kahraman [1], Yongping Li[1], Nadja Grammes[1], Lars Geffers [3], Christina Backes [1], Rudi Balling [3,4,5], Fabian Kern[1], Rejko Krüger[3,4,5], Frank Lammert [6], Nicole Ludwig[7], Benjamin Meder[8], Bastian Fromm [9], Walter Maetzler[10], Daniela Berg[10], Kathrin Brockmann[11], Christian Deuschle[11], Anna-Katharina von Thaler [11], Gerhard W. Eschweiler[12], Sofiya Milman[13], Nir Barziliai[13], Matthias Reichert [6], Tony Wyss-Coray [2], Eckart Meese[7] & Andreas Keller [1,2,14 ✉]

Aging is a key risk factor for chronic diseases of the elderly. MicroRNAs regulate post-transcriptional gene silencing through base-pair binding on their target mRNAs. We identified nonlinear changes in age-related microRNAs by analyzing whole blood from 1334 healthy individuals. We observed a larger influence of the age as compared to the sex and provide evidence for a shift to the 5' mature form of miRNAs in healthy aging. The addition of 3059 diseased patients uncovered pan-disease and disease-specific alterations in aging profiles. Disease biomarker sets for all diseases were different between young and old patients. Computational deconvolution of whole-blood miRNAs into blood cell types suggests that cell intrinsic gene expression changes may impart greater significance than cell abundance changes to the whole blood miRNA profile. Altogether, these data provide a foundation for understanding the relationship between healthy aging and disease, and for the development of age-specific disease biomarkers.

[1] Chair for Clinical Bioinformatics, Saarland University, 66123 Saarbrücken, Germany. [2] Department of Neurology and Neurological Sciences, Stanford University, Stanford, CA 94305, USA. [3] Luxembourg Center for Systems Biomedicine, 4362 Esch-sur-Alzette, Luxemburg. [4] Transversal Translational Medicine, Luxembourg Institute of Health (LIH), 1445 Strassen, Luxemburg. [5] Parkinson Research Clinic, Centre Hospitalier de Luxembourg, 1210 Luxembourg, Luxemburg. [6] Internal Medicine, Saarland University, 66421 Homburg, Germany. [7] Human Genetics, Saarland University, 66421 Homburg, Germany. [8] Internal Medicine, University Hospital Heidelberg, 69120 Heidelberg, Germany. [9] Department of Molecular Biosciences, Stockholm University, 11418 Stockholm, Sweden. [10] Department of Neurology, Christian-Albrechts-Universität zu Kiel, 24105 Kiel, Germany. [11] TREND study center Tübingen, Tübingen, Germany. [12] Geriatric Center and the Department of Psychiatry and Psychotherapy, University Hospital Tübingen, 72076 Tübingen, Germany. [13] The Institute for Aging Research, Albert Einstein College of Medicine, New York, NY 10461, USA. [14] Center for Bioinformatics, Saarland Informatics Campus, Saarland University, 66123 Saarbrücken, Germany. ✉email: andreas.keller@ccb.uni-saarland.de

A ging is the leading risk factor for cardiovascular disease, diabetes, dementias including Alzheimer's disease, and cancer, together accounting for the majority of debilitating illnesses worldwide[1]. Uncovering common therapeutic targets to prevent or treat these diseases simultaneously could convey enormous benefits to quality of life. It is therefore essential to model the cellular processes culminating in these diverse maladies through an understanding of the molecular changes underlying healthy and pathological aging[2]. Accordingly, a variety of molecular studies have been conducted in humans, including whole genome analysis of long-lived individuals[3], transcriptomic analyses of tissues[4], plasma proteomic profiling[5], and the exploration of epigenetic control of aging clocks[6]. Recent organism-wide RNA-sequencing data of whole organs and single cells across the mouse lifespan provide an important and complementary database from which to build models of molecular cascades in aging[7,8].

Functional improvement of aged tissues has been achieved by an expanding number of techniques, ranging from dietary restriction[9] to senescent cell elimination and partial cellular reprogramming. This also includes heterochronic parabiosis, in which an old mouse is exposed to a young circulatory system. These experiments point to systemic factors in the blood of young mice that modulate organ function in aged animals[10,11]. Indeed, the list of individual plasma proteins with beneficial or detrimental effects on different tissues is growing. It is likely, however, that each plasma protein interacts with complex intracellular regulatory networks, and that alterations to such networks are a key component of aging and rejuvenation.

Non-coding ribonucleic acids like microRNAs (miRNAs) represent essential players governing these molecular cascades, and they show a highly complex spectrum of biological actions[12–14]. MicroRNAs are a family of short single stranded non-coding RNA molecules that regulate post-transcriptional gene silencing through base-pair binding on their target mRNAs[13], thereby regulating most if not all cellular and biological processes[15]. Yet, their involvement in the aging process and rejuvenation of aged tissues is often ignored by transcriptomic studies and is thus largely uncharacterized. A single microRNA targets not only untranslated regions (UTRs) of numerous genes, but it can also bind multiple sites within a single UTR[16]. Similarly, a UTR of a specific gene can contain target sites for dozens or even hundreds of miRNAs. Since their discovery, miRNA changes have been reported for almost all cancers and many non-cancer diseases like Alzheimer's disease[17,18], multiple sclerosis[19], or heart failure[20]. And although relatively sparse, several studies have measured aging miRNA expression in different human and primate tissues[21]. For example, Somel and co-workers analyzed miRNA, mRNA, and protein expression linked to development and aging in the prefrontal cortex of humans and rhesus macaques over the lifespan[22]. Likewise, changes of miRNA levels in aging human skeletal muscle have been characterized[23], as have miRNA levels in body fluids such as serum[24,25]. In whole blood, we previously reported a significant number of age-related miRNAs[26], and Huan and co-workers measured a selection of miRNAs by RT-qPCR in whole blood from over 5000 individuals from the Framingham Heart Study[27]. While these initial studies are intriguing, they can be limited by the use of discrete time points, incomplete lifespan coverage, limited cohort sizes, and incomplete miRNA panels.

Here, we performed a comprehensive characterization of all 2549 annotated miRNAs (miRBase V21) in 4393 whole blood samples from both sexes across the lifespan (30–90 years). To understand the relationship between healthy aging and disease, we included 1334 healthy controls (HC), 944 patients with Parkinson's disease (PD), 607 with heart diseases (HD), 586 with non-tumor lung diseases (NTLD), 517 with lung cancer

(LC), and 405 with other diseases (OD) (Fig. 1a, b; Supplementary Data 1).

## Results

**miRNA profiles are stronger associated with the age as compared to the sex**. We first sought to model healthy aging as a baseline for understanding disease. As males have shorter lifespans than females, and each sex suffers a different array of age-related diseases, we investigated the interplay between age and sex on blood miRNA profiles. Confirming our previous observation in a cohort of 109 individuals[26], we found that age has a more pronounced influence than sex. In fact, 1568 miRNAs significantly correlated with age, but only 362 correlated with sex according to Benjamini–Hochberg adjusted p-values of the Wilcoxon Mann–Whitney test (Fig. 2a, b). While 231 miRNAs overlapped between these groups, this number was not significant (two-sided Fisher's exact test $p$-value of 0.35; Pearson's Chi-squared Test of 0.36), suggesting that, in general, those miRNAs changing with age are shared by both sexes, and those specific to one sex do not change with age. In consequence, the Spearman correlation coefficient (SC) of age-related changes between males and females was high (SC of 0.884, $p < 10^{-16}$, Fig. 2c).

We next sorted miRNAs by their correlation with age, regardless of their significance, and assigned each to one of 5 groups: strongly decreasing with age (cluster 1: 174 miRNAs, SC < −0.2), moderately decreasing (cluster 2: 382 miRNAs; −0.2 < SC < −0.1), unaltered (cluster 3: 1451 miRNAs; −0.1 < SC < 0.1), moderately increasing (cluster 4: 368 miRNAs; 0.1 < SC < 0.2), and strongly increasing (cluster 5: 174 miRNAs, SC > 0.2) (Supplementary Data 2). As miRNAs regulate a diverse array of critical pathways[28], we performed microRNA enrichment analysis and annotation (miEAA) on this sorted list, thereby calculating a running sum of miRNAs associated with each of ~14,000 biochemical categories and pathways. We revealed a remarkable disequilibrium between the number of pathways related to downregulated miRNAs (76 pathways) and upregulated miRNAs (620 pathways; adjusted $p$-value < 0.05; Supplementary Data 3). This is even more striking considering the number of miRNAs increasing or decreasing did not differ significantly (556 with SC < −0.1; 542 with SC > 0.1), and suggests that miRNAs increasing with age have a higher functional relevance. Reassuringly, for miRNAs decreasing with age we found "Negative Correlated with Age" ($p = 4 \times 10^{-10}$) among the most significant categories (Fig. 2d). A large fraction of the top pathways regardless of the miRNA direction were enriched for brain function and neurodegeneration, including "Downregulated in Alzheimer's Disease" ($p = 10^{-5}$), "regulation of synaptic transmission" ($p = 0.028$), and "APP catabolic processes" ($p = 0.032$) (Fig. 2e, Supplementary Fig. 1a–l).

Although such linear correlation analyses can reveal meaningful biological features, the importance of nonlinear aging changes, such as those found for plasma proteins[5] and tissue gene expression, is becoming increasingly evident. We therefore aimed to use the high temporal resolution of the dataset to more thoroughly understand whole blood miRNA dynamics across the lifespan. We first plotted miRNA trajectories for each of the 5 clusters (Supplementary Fig. 2), confirming many miRNAs exhibit non-linear patterns. By comparing linear and nonlinear correlations for each, we uncovered nonlinear changes in 116 of the 1098 miRNAs altered with age, of which 90 decreased and 26 increased (Fig. 2f, g, Supplementary Data 4). A miEAA analysis highlighted a significant enrichment of miRNAs following nonlinear trajectories with aging in basically all human tissues[29] (Fig. 2h). This finding stands out considering the high degree of tissue specificity of miRNAs. We thus speculate that diseases

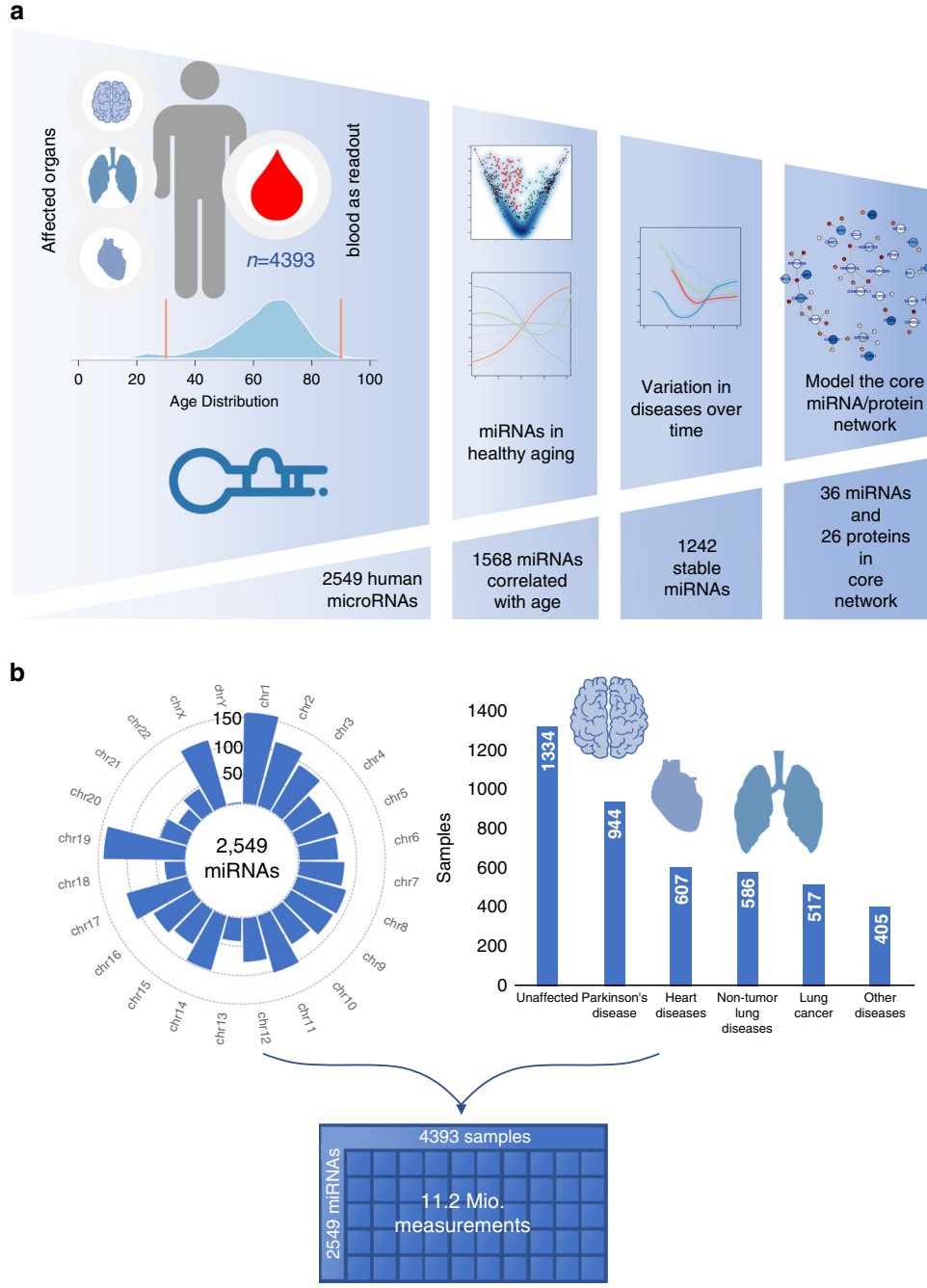

**Fig. 1 Study characteristics. a** Study set up and analysis workflow from high-throughput data to a specific aging network. The cohort consist of 4393 samples of which the age distribution is provided. For the 4393 samples genome wide miRNA screening using microarrays has been performed. The first analysis describes 1568 miRNAs that are correlated to age in healthy individuals. In the second step we identified disease specific miRNA changes with aging and finally define a set of 1242 miRNAs that are not affected by diseases. Finally, to model regulatory cascades in healthy aging we related the miRNA data to plasma proteins and identified a core aging network. **b** The circular plot shows the genome wide nature of our miRNA approach, all miRNAs from miRBase V21 were included in the experimental analysis. We measured 4393 samples for the abundance of these miRNAs, resulting in a 2549 times 4393 data table containing 11.2 million miRNA measurements that correspond to over $2 \times 10^8$ spots on the arrays.

affecting these organs might be associated with changes in blood miRNA profiles.

**miRNA arm shifts are associated with aging**. A shift in the expression of the 3' and 5' mature arm of miRNAs is observed between different tissues[30] tissues but also in healthy and diseased conditions such as cancer[31]. We speculated that likewise aging may affect the arm distribution and searched for respective

arm shift events. Indeed, we observed a correlation of the arm specific expression in 40 cases (Supplementary Data 5). For 27 miRNAs (67.5%) we observed increasing 5' mature expression and decreasing 3' expression over age while in 13 cases 32.5% of cases the 3' form increased and the 5' form decreased. These results indicate a generally increasing 5' mature miRNA expression with aging. The largest absolute increase of 5' mature expression was identified for miR-6786. A miRSwitch analysis highlighted that usually the 3' form is dominating in H. sapiens

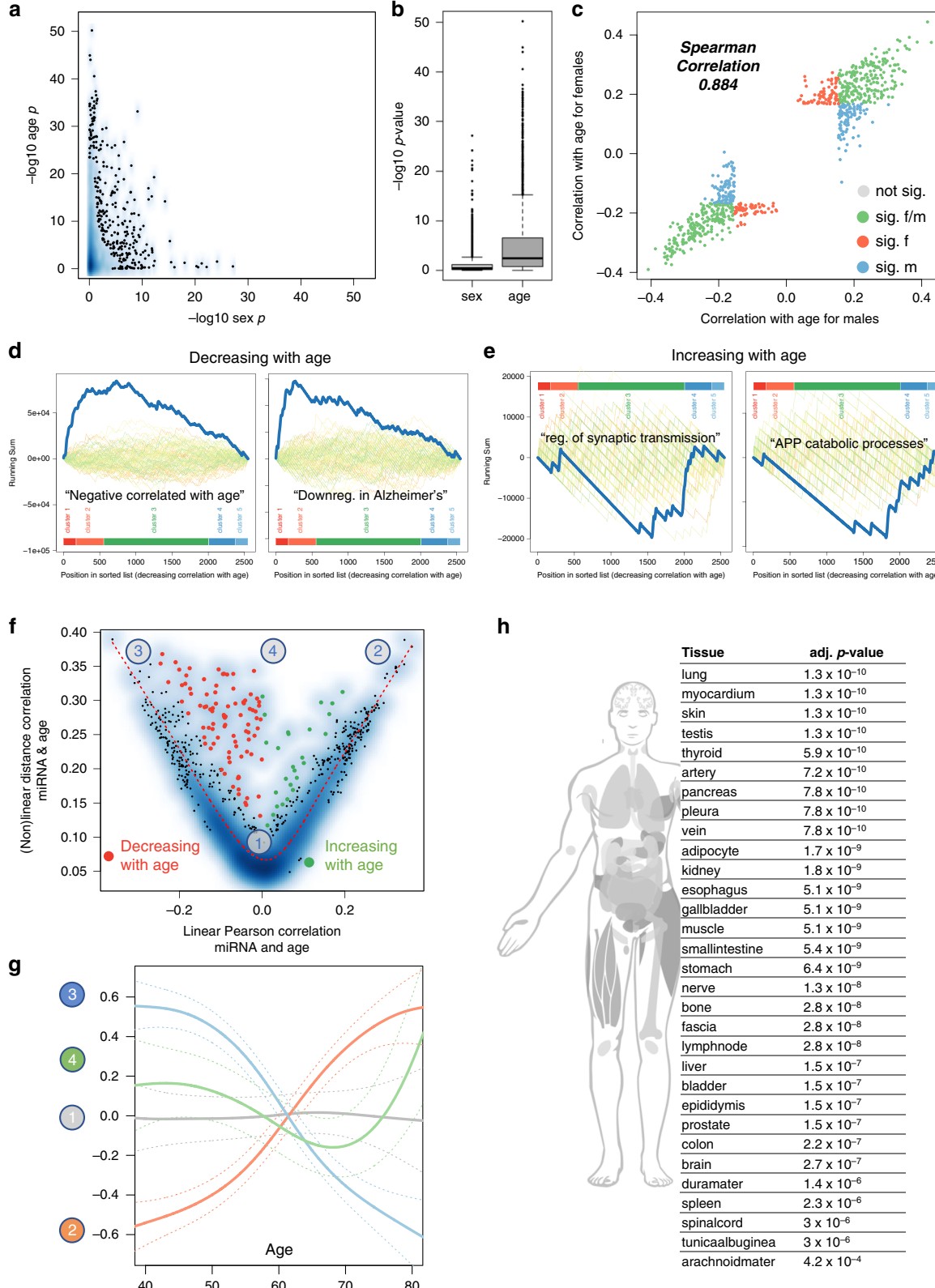

with 5' dominance mostly in plasma samples. For the miRNA with the most decreasing 5' expression ratio (miR-4423) we found dominating 3' expression mostly in breast milk, the heart, testis, stem cells and blood cells. Our results thus suggest an altered ratio of the 3' to 5' mature expression ratio that might be attributed to or effect different tissues.

**The association between age and miRNA expression is partially lost in diseases**. Although the cellular and molecular degeneration of aging often instigates age-related disease, there are nonetheless elderly individuals who have lived entirely disease-free lives. We therefore asked what differentiates such healthy aging from aging resulting in disease. For each disease and healthy controls, we

**Fig. 2 miRNAs dependency on age and gender. a** Smoothed scatter plot of the two-tailed age and gender association p-value for 2549 miRNAs. P-values for the sex are computed using Wilcoxon Mann–Whitney test and for the Spearman Correlation via the asymptotic t approximation. The p-values are Benjamini–Hochberg adjusted. **b** Boxplot of the age and gender p-value from **a** for 2549 miRNAs. The box spans the 25% and 75% quantile, the solid horizontal line represents the median and the whiskers extend to the most extreme data point which is no more than 1.5 times the interquartile range from the box. **c** Correlation of miRNAs with age in males and females. Gray dots: not significant; orange and blue dots: miRNAs significantly correlated with age only in males or females; green dots: miRNAs significantly correlated with age in males and females. **d** Results of the miRNA enrichment analysis. Colored curves in the background represent random permutations of miRNAs. The cluster membership is projected next to the order of miRNAs. The category "negative correlated with age" is highly significant and confirms our data in general. Also, the category "downregulated in AD" is enriched with miRNAs decreasing over age. **e** Regulation of synaptic transmission is among the categories being enriched in miRNAs going up with age. Moreover, APP catabolic processes is another category being enriched in miRNAs going up with age. **f** Linear Pearson correlation versus non-linear distance correlation for the association of age to miRNAs. Orange dots have a high non-linear correlation that is not explained by linear correlation and are decreasing with age, green dots have a high non-linear correlation that is not explained by linear correlation and are increasing with. The orange dotted line represents a smoothed spline and the four numbers in gray circles represent the position of miRNAs where examples are provided in **g**. **g** Examples of correlation for miRNAs with age. (1) gray: no correlation; (2) orange dominantly positive linear correlation; (3) blue dominantly negative linear correlation; (4) non-linear correlation. Each solid line is a smoothing spline. **h** Tissue enrichment for the miRNAs that are correlated with age in a non-linear fashion. The human model has all organs highlighted in gray that are significantly enriched. The table on the right lists the organs with corresponding p-values. P-values have been computed using the hypergeometric distribution and were adjusted for multiple testing using the Benjamini–Hochberg approach.

computed the Spearman correlation (SC) with age for all 2549 miRNAs (Fig. 3a, Supplementary Data 6). Overall, healthy controls reached the largest absolute SC, greater than twice that of the pooled disease cohort, and larger than any individual disease. Using an Analysis of variance, we found highly significant differences ($p < 2.2 \times 10^{-16}$) and a non-parametric Wilcoxon Mann–Whitney test confirmed the significant differences of absolute Spearman correlation in healthy versus diseased samples ($p < 2.2 \times 10^{-16}$). In line with these findings, samples from healthy individuals showed far more miRNAs with significant age correlations (Fig. 3b), suggesting that the presence of an age-related disease may disrupt healthy aging miRNA profiles (Wilcoxon Mann–Whitney test $p < 2.2 \times 10^{-16}$). For example, lung cancer patients were enriched for a positive correlation with age, while miRNAs in patients with heart disease were enriched for negative correlation with age. We then compared the miRNA trajectories from the 5 clusters of healthy individuals to the matched clusters in diseased patients (Supplementary Fig. 2), and similarly, miRNAs from diseased individuals show far weaker aging patterns. This held true both when each disease was analyzed separately, or pooled.

To determine the extent to which diseases affect miRNA abundance compared to healthy controls, we computed the number of differentially expressed miRNAs between cases and controls using a sliding window analysis. That is, we first compared diseased individuals aged 30–39 years to healthy individuals aged 30–39 years, then increased the window in increments of one year (31–40 years, 32–41 years, etc.) to the final window of 70–79 years (Fig. 3c, Supplementary Fig. 3a, b). As the age distribution varied between these groups, we excluded any window in which there were fewer than 20 disease cases and 20 healthy controls. Interestingly, for all diseases the number of differentially expressed miRNAs was high in young adults but decreased sharply into middle age, plateauing around age 60 for lung cancer and 50 for non-tumor lung diseases. Heart diseases largely plateaued by the early 50s. Parkinson's disease (PD), on the other hand, reached a minimum around age 47 before sharply increasing. With the exception of PD, these data show that aged healthy and diseased individuals are more similar than younger healthy and diseased individuals, perhaps suggesting that aged healthy individuals share some phenotypic characteristics of heart and lung disease.

We next asked if these diseases shared any miRNA alterations, and surprisingly we found that those miRNAs most commonly dysregulated were also those with the largest effect size (Fig. 3d). These pan-disease miRNAs included miR-191-5p (Fig. 3e), which targets mRNAs involved in cellular senescence[28]. We also observed disease-specific miRNAs like miR-16-5p, which targets the PI3K-Akt signaling pathway and microRNAs involved in lung

cancer[28]. In summary, miRNA expression seems to be orchestrated in healthy aging with a loss of regulation in disease. In addition to disease-specific miRNAs, there appears to be a group of pan-disease miRNAs that change in a distinct manner. We thus asked on the specificity of biomarkers for diseases, especially in an age dependent context.

**Distinct miRNA biomarker sets exist in young and old patients**. The previous analyses of biomarkers in diseases were largely quantitative, i.e., we computed the number of dysregulated miRNAs in diseases for young and old patients. Here, we set to evaluate changes in the miRNA sets for young and old patients in the diseases. In this context we made use of the dimension reduction and visualization capabilities of self-organizing maps (SOMs). First, we considered the effect sizes of miRNAs for the two most global comparisons, i.e., healthy controls versus diseases and old (60–79 years) versus young (30–59 years) individuals. The heat map representation for the healthy versus disease comparison (Fig. 4a) and for young versus old individuals (Fig. 4b) highlights distinct patterns for the two comparisons and indicates that the aging miRNAs are different from the general disease miRNAs. This analysis however calls for a disease specific consideration. To this end we computed for each of the four diseases biomarkers in old and young patients using again the effect size as performance indicator and the self-organizing map analysis followed by a hierarchical clustering (Fig. 4c). While the cluster heat maps identify larger differences between the disease biomarker sets as compared to young and old biomarkers, also the sets within the diseases vary greatly (Fig. 4c). In line with the previous analyses we observe larger effects for all diseases but PD in young patients (middle row of Fig. 4c). In old patients, the respective biomarkers are partially lost. Only in few cases new biomarkers emerge in old patients that are not present in young patients. As the full annotation of the SOM grid shows, each SOM cell has an average of 8 cluster members with a standard deviation of 3.5 miRNAs (Supplementary Data 7). The distribution largely corresponds to a normal distribution, only four cells (24, 62, 81, and 82 in Supplementary Data 7) contain more than 15 miRNAs (mean + two times the standard deviation).

The previous analyses suggest distinct biomarker sets for young and old patients in the different diseases. As a consequence, future biomarker test based on miRNAs may not only be established for a disease but for a specific age range of patients with that disease.

Given the results from this and the previous section we computed for each miRNA in each disease and each age window

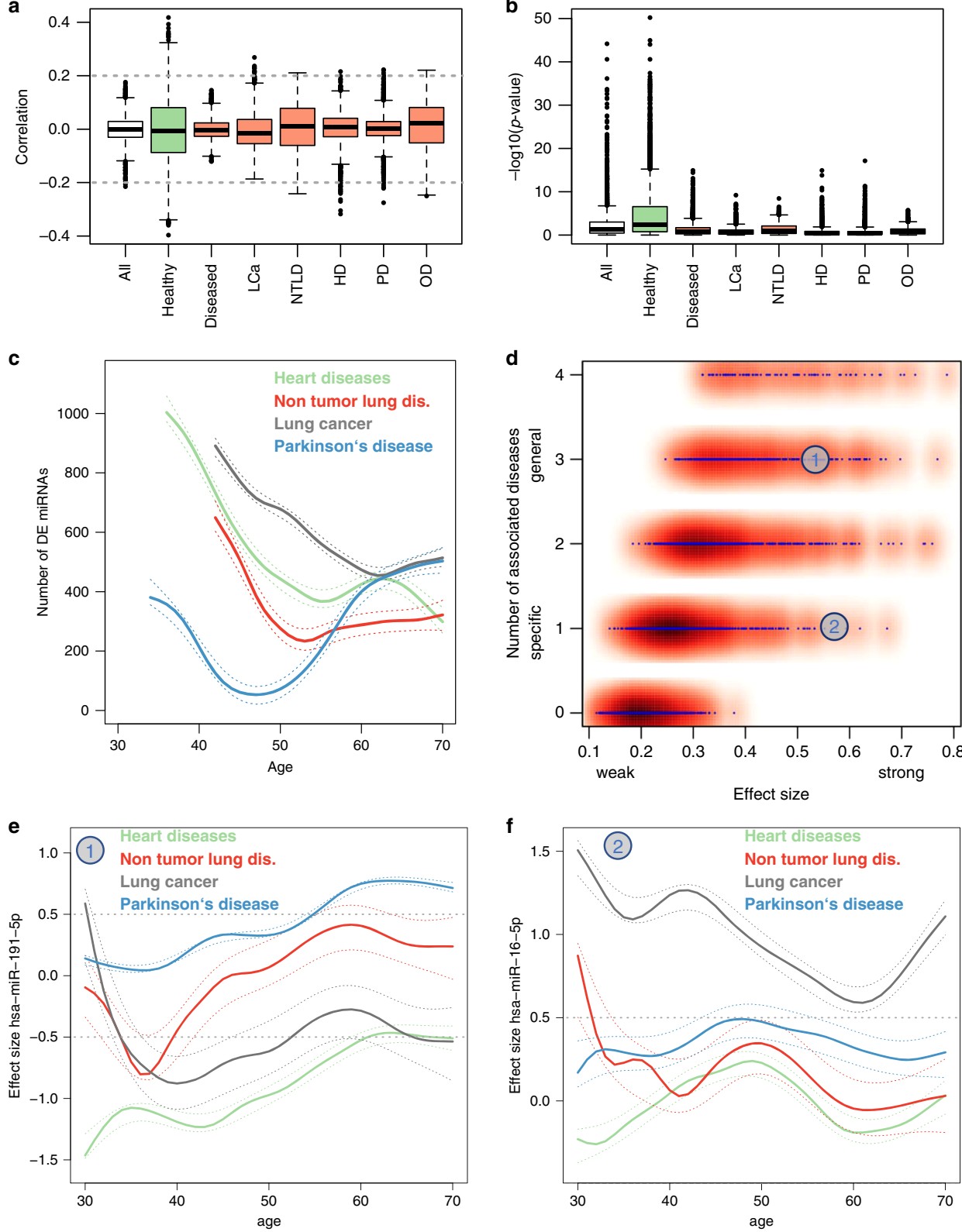

the effect size (Supplementary Data 8). The respective supplementary data provides detailed insights in how specific certain miRNAs are for specific diseases and age ranges and can support ongoing biomarker studies significantly.

All results obtained so far argue for a strong immunological component of the miRNAs, and as a consequence of miRNA target networks. Since our experimental system profiles whole blood miRNAs, we set out to determine the cellular origin by computational deconvolution.

**White blood cells are the major repository of miRNAs in whole blood.** Circulating immune cells have been implicated in aging and a variety of age-related diseases, and one of the most

**Fig. 3 Diseases miRNAs are affected by age effects. a** Boxplot of the Spearman correlation coefficient for each miRNA to all samples, healthy individuals, and patients. Group sizes: $n_{HC} = 1334$, $n_{PD} = 944$, $n_{HD} = 607$, $n_{NTLD}$, $n_{LC} = 517$, $n_{OD} = 405$. The box spans the 25% and 75% quantile, the solid horizontal line represents the median and the whiskers extend to the most extreme data point which is no more than 1.5 times the interquartile range from the box. **b** Boxplot of $p$-values for the Spearman correlation coefficient of each miRNA to all samples, healthy individuals, and patients from **a**. Group sizes: $n_{HC} = 1334$, $n_{PD} = 944$, $n_{HD} = 607$, $n_{NTLD}$, $n_{LC} = 517$, $n_{OD} = 405$. The box spans the 25% and 75% quantile, the solid horizontal line represents the median and the whiskers extend to the most extreme data point which is no more than 1.5 times the interquartile range from the box. The $p$-values have been computed via the asymptotic t approximation. **c** Number of deregulated miRNAs in disease groups depending on different ages in a sliding window analysis. Each solid line is a smoothing spline (green–heart diseases; red–non tumor lung diseases; gray–lung cancer; blue–Parkinson's disease). The areas represent the 95% confidence intervals. For all disease groups, the number of deregulated miRNAs decreases with age while it increases for Parkinson's Disease. **d** Smoothed scatterplot showing the average effect size per miRNA dependent on the number of diseases where the miRNA is associated with. In the lower right corner (the $y$-axis value of 1) the specific miRNAs with high effect sizes can be found. In the upper right corner, miRNAs with high effect sizes independent of the disease are located. The two numbers represent the location of the examples provided in **e** and **f**. **e** Example of a miRNA that is downregulated in heart diseases of younger patients, upregulated in older Parkinson's patients and not deregulated in lung diseases. Each solid line is a smoothing spline (green–heart diseases; red–non tumor lung diseases; gray–lung cancer; blue–Parkinson's disease). The areas represent the 95% confidence intervals. **f** Example of a miRNA from the lower right part of Fig. 3d. The miRNA is significant upregulated in lung cancer independent of age but basically not associated with other diseases. Color codes of panels **c**, **e**, and **f** are matched. Each solid line is a smoothing spline (green–heart diseases; red–non tumor lung diseases; gray–lung cancer; blue–Parkinson's disease). The areas represent the 95% confidence intervals.

common diagnostic tests for disease is blood cell profiling. Since miRNAs are known to be enriched in different blood cell types[32], we performed computational deconvolution of the whole blood miRNA profile, thereby grouping miRNAs by their predicted cell type(s) of origin (Fig. 5a). A total of 196 miRNAs were attributed to one specific cell type, including 127 miRNAs arising from monocytes. Most others derive from three or more types. For example, the largest group of 139 miRNAs stems from a combination of white and red blood cells (WBCs, RBCs), exosomes, and serum. And the third largest group of 119 is restricted to six types of WBCs. We also observed 31 miRNAs specific for NK cells, 19 specific for T-helper cells, 11 specific for B cells, and 8 specific for cytotoxic T cells. Overall, for those miRNAs for which we could assign a prospective origin, we found WBCs as the main contributor, even though they represent a substantially smaller volume of whole blood relative to RBCs and serum (Fig. 5b).

We then applied this analysis to those miRNAs changing with age, and found that those increasing appear to largely originate from B cells, monocytes, NK cells, cytotoxic T cells, and serum (Fig. 5c). In contrast, miRNAs decreasing with age are those enriched in neutrophils, T helper cells, and RBCs. These data indicate shifts in aging miRNA trajectories of specific blood cell types (Supplementary Fig. 4). Interestingly, for the above cell types, known age-related abundance changes largely follow opposite trends: lymphocytes generally decrease with age while neutrophils increase with age[33]. This suggests that cell-intrinsic gene expression changes age may significantly contribute to the observed whole blood miRNA profiles.

**miRNAs associated with healthy aging regulate the expression of plasma proteins**. An increasing body of evidence points to functional roles of systemic plasma proteins in aging and disease[5]. These proteins may represent downstream targets of blood-borne miRNAs. We thus compared our data to a recent dataset of plasma proteins associated with age in healthy individuals[5]. Because miRNAs regulate genes/proteins in a complex network, miRNAs increasing with age do not necessarily lead to downregulation of all target genes/proteins, and vice versa. Accordingly, we observed only one tendency: miRNAs decreasing with age (cluster 1 and 2) showed a slight enrichment for regulating proteins increasing with age (Fig. 6a). Considering such complexity, we employed a network-based analysis. Using all pairwise interactions of miRNAs with plasma proteins, we first computed a regulatory network (Fig. 6b). From this, we extracted a core network containing the top 5% downregulated miRNAs

and the top 5% upregulated proteins, which was then further refined by including only experimentally validated miRNA/target genes mined from the literature[34], as well as miRNA/target pairs with an absolute Spearman correlation of at least 0.6. This stringent core network consists of 36 miRNAs targeting 26 genes (proteins) and splits into two larger and six smaller connected components (Fig. 6c). The densest part of the core network contains the axon guidance related semaphorin 3E (SEMA3E) and serine and arginine rich splicing factor 7 (SRSF7), which were targeted by 8 miRNAs including miR-6812-3p (Fig. 6d, Supplementary Fig. 5, Supplementary Fig. 6). Intriguingly, there exist no studies of this miRNA, but it targets SEMA3E in an age dependent manner with a Spearman correlation of −0.89.

Finally, we investigated the possible cell type of origin of these core miRNAs with deconvolution, which showed enrichment for neutrophils, monocytes, and B cells (Fig. 6e). We then used single-cell PBMC transcriptomic data to determine if SEMA3A or SRSF7 were expressed in these same cell types. While SEMA3E was not detectable, we did observe SRSF7 expression widely across cell types, including neutrophils, monocytes, and B cells (Fig. 6f, g). SRSF7 plays a role in alternative RNA processing and mRNA export, but has no known role in aging or neurodegeneration. Further research will be required to determine if miRNAs like miR-6812-3p do indeed target SRSF7 in these specific cell types, and to uncover if this process contributes to the global decline of transcription observed with age.

## Discussion

Our analysis of blood derived microRNAs provides insights into changes in microRNA abundance dependent on age, sex, and disease. While age clearly contributes to expression changes, sex has a more modest effect. In fact, most miRNAs show a similar behavior over the lifespan in males and females. This is generally in-line with recent results in transcriptomic mouse tissue aging[7,8]. Generally, our results compare well to other studies of miRNAs in aging[27], especially regarding miRNAs increasing with age, for which we observe high concordance. There are, however, miRNAs decreasing with age reported in the previous study for which we did not find evidence. The most extreme examples are miR-30d-5p and miR-505-5p, both increasing with age in our study in the healthy individuals. Nonetheless, given different cohorts with different ethnicity, varying age range, and distinct profiling technologies, we observed remarkable concordance between the studies.

Here, we observed that diseases globally disturb the normal aging progression of blood-borne miRNAs. While linear

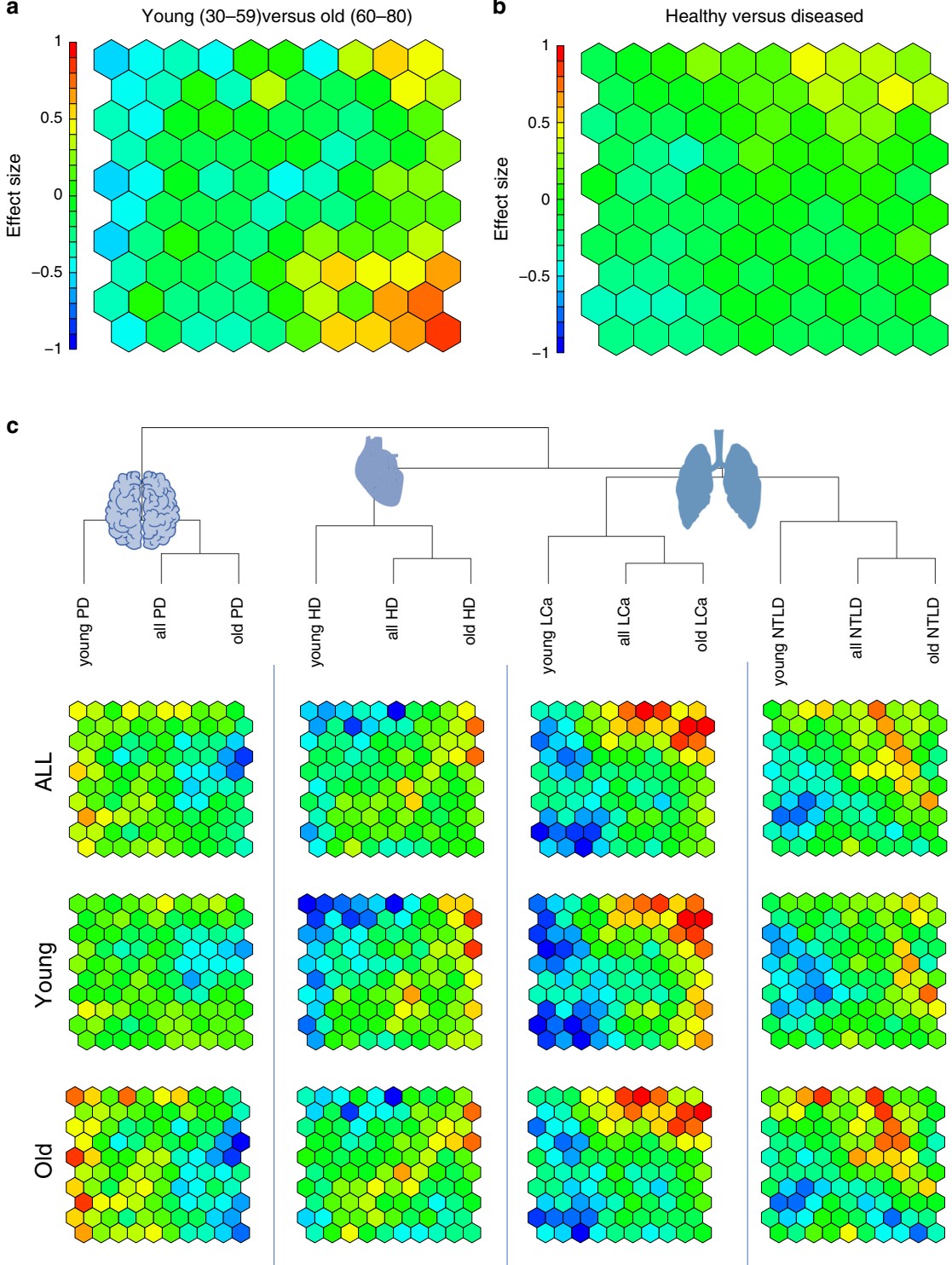

**Fig. 4 Disease specificity of miRNA biomarkers. a** Heat map representation of the SOM analysis as a 10 × 10 grid with 100 entries. Each cell contains at least one miRNA and up to 20 miRNAs. The full annotation of miRNAs to cells are provided in Supplementary Data 7). The cells are colored by the effect size of miRNAs for the comparison in old versus young. Red cells contain miRNAs with effect sizes >0.5 that are upregulated and in blue miRNAs that are downregulated with effect sizes <−0.5. **b** Same heat map as in **a** but colored for the difference in young versus old. The scale for the effect size has been kept the same as **a**. Thus fewer yellow/red, as well as blue spots indicate overall lower effect sizes. **c** Clustering of the SOM results in biomarkers for the four diseases and in all biomarkers independently of age, biomarkers for young patients and biomarker for old patients. The dendrogram has been computed from hierarchical clustering (complete linkage on the Euclidean distance). In all cases the biomarkers cluster by disease and not by age and the old biomarker set is closest to the all biomarker set while the young biomarker set has larger distances. Overall, NTLD and LCa markers are closest to each other, second closest are heart biomarkers and most different PD biomarkers. The SOM cells clearly highlight differences between biomarkers for diseases in young and old patients.

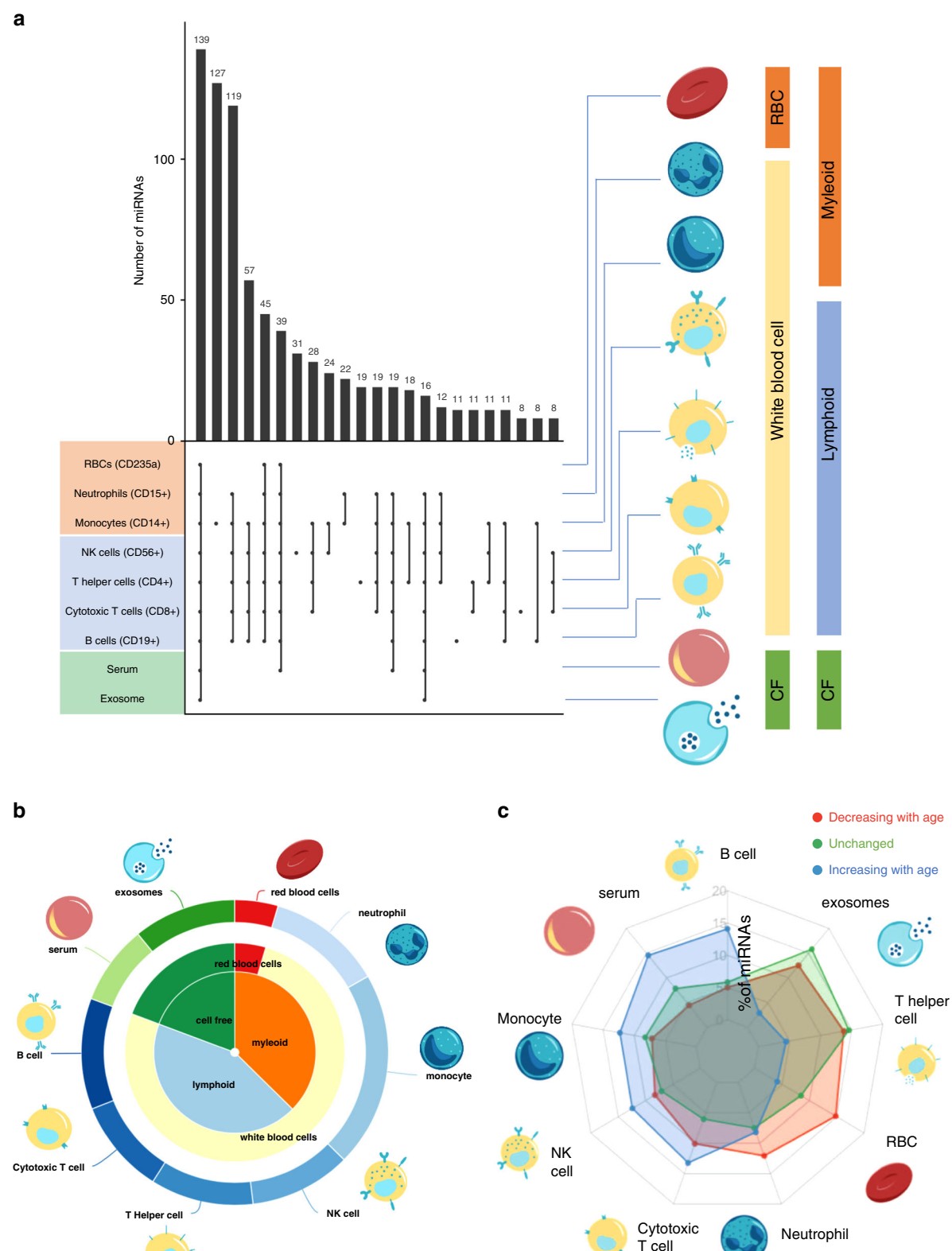

**Fig. 5 Blood cell deconvolution. a** The distribution of miRNAs in the different blood compounds. The rows are sorted by the blood compounds given on the right (RBC: red blood cell; CF: cell free), the columns are sorted according to a decreasing number of miRNAs. **b** Relative abundance of all miRNAs in the different blood compounds. **c** Distribution of miRNAs in cell types. The green distribution is the background and presents the relative composition of 1451 miRNAs in cluster 3. The blue distribution represents miRNAs increasing by age (cluster 4&5) and are enriched e.g., in B cells and serum. The red distribution represents miRNAs decreasing by age (cluster 1&2) and are enriched e.g., in neutrophils and RBCs.

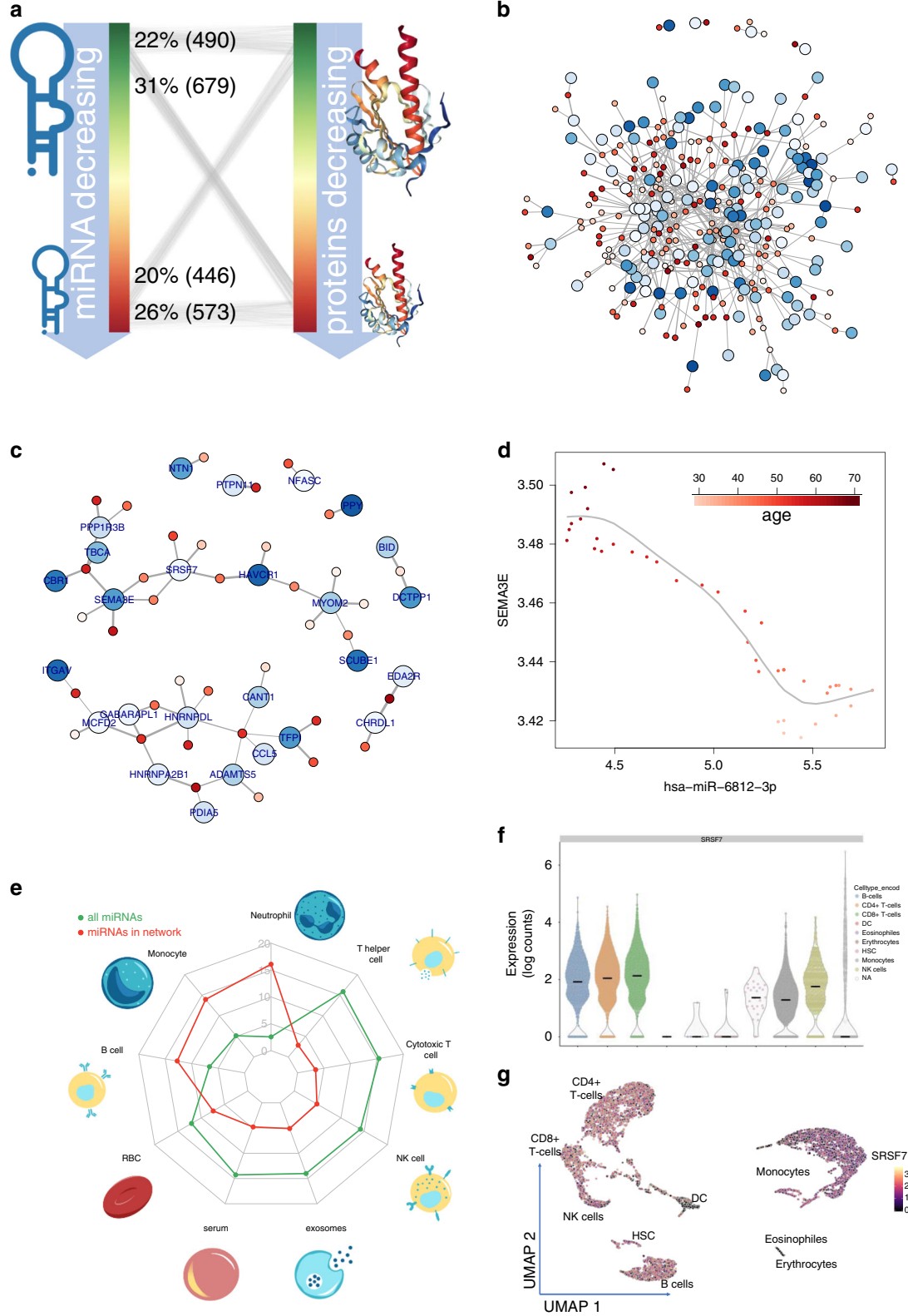

modeling insufficiently explained changes with aging, distance correlation analysis identified 90 miRNAs that were decreasing and 26 that were increasing with age in a non-linear manner. These effects are, however, frequently not disease specific. If disease specific effects occur, they appear to establish themselves in given time windows throughout live. For example, lung and heart diseases show the largest effect sizes in the 4th to 5th decade

of life, and Parkinson's disease showed the largest effect size in the 6th to 7th decade. All known biological factors including age, sex, and disease status together only explained part of the overall data variance. Thus, unknown biological variables and technical factors also contribute to miRNA abundance.

Our results underline not only the importance of age as a confounder in biomarker studies, but they show that age needs to be

**Fig. 6 Age related miRNAs are correlated to age related proteins. a** Correlation of miRNAs to proteins. miRNAs and proteins are sorted by increasing correlation with age. Thin lines are miRNA/gene interactions between top/bottom 10% of miRNAs and proteins. Numbers represent actual count of edges. **b**, **c** Core network. Proteins (larger nodes) are targeted by miRNAs (smaller nodes). Edge width correspond to the correlation. Blue nodes represent increase with age, red nodes decrease with age. The outer circles of the protein nodes indicate an expected an influence of the miRNAs leading to an increase with age. Panel **c** represents a more stringent version of the network from panel **b**. **d** One representative example of an edge from the network in **b**, **c**: SEMA3E and miR-6812-3p. Each dot represents all individuals in a time interval of 10 years, shifted between 30 and 70 years. SEMA3E is high expressed in older individuals while miR-6812-3p is low expressed (dark red points in the upper right corner). In young individuals the pattern is opposite (tale points in the lower right corner). **e** Blood cell compound distribution. miRNAs from the core network come from neutrophils, monocytes and B cells. **f** Violin plot of expression of SRSF7 in human blood cells. **g** UMAP embedding of human blood cells colored by expression of SRSF7.

incorporated into the definition of disease biomarkers. The age dependency of miRNA biomarkers may be even more prominent for acute diseases that are accompanied by drastic molecular changes. Furthermore, the influence of a disease on healthy aging miRNA patterns suggests that it is conceivable to define "negative biomarkers", i.e., biomarkers that reflect the degree of disturbance of a given time-dependent pattern typically found in healthy individuals.

miRNAs comprise complex gene regulatory networks, and it is essential to identify the miRNA-targets that are regulated by a given miRNA network. However, this is already a demanding task for static networks, and it becomes even more challenging when considering how entire networks change with age. We attempted to overcome this complexity and identify a core miRNA network by implementing several stringent criteria: (i) the inclusion of miRNA-gene pairs only if experimental evidence exists, (ii) limiting the analysis to the top 5% of miRNAs decreasing with age, and (iii) the top 5% of proteins increasing with age and with pairwise absolute correlation of at least 0.6. This stringent parameter set identified a core network of 36 miRNAs and 26 proteins organized in two larger hubs with eight miRNAs targeting the axon guidance related semaphorin 3E (SEMA3E) and serine and arginine rich splicing factor 7 (SRSF7). Semaphorines play crucial roles during the development of the nervous system, especially in the hippocampal formation[35]. SEMA3E suppresses endothelial cell proliferation and angiogenic capacity, and in complex with PlexinD1 it inhibits recruitment of pericytes in endothelial cells[36]. Since we did not detect SEMA3E mRNA expression in single blood cell data we also explored other sources such as the Genotype-Tissue Expression (GTEx) project[37]. But also in the GTEx data no expression for the gene was reported in bulk sequencing data. It thus remains unclear how or if these miRNAs directly or indirectly impact SEMA3E protein levels in plasma. In this context, low abundant fractions of the blood such as exosomes might play a role. However, SRSF7, which belongs to a protein family linking alternative RNA processing to mRNA export[38], is expressed across a variety of circulating immune cells. This is intriguing as no role in aging or neurodegeneration is known.

Often, different technologies are available for high-throughput studies. To characterize the complete miRNome, usually microarrays or high-throughput sequencing are used. The choice of the best technology depends both, on technical factors and on the underlying biological question to be addressed. We decided to use microarray technology mostly because of the high dynamic range of blood miRNAs. In whole blood, the majority of reads (90–95%) are matching to few (2–5) miRNAs[39]. While generally a depletion is feasible[40], it bears the risk to alter the profile of other miRNAs especially since it has to be tailored for the respective sequencing technology. To use microarrays has however also disadvantages. MicroRNAs are often modified and build so-called isomiRs and basically all human miRNAs express different isoforms[41]. Likewise, data from the Rigoutsos lab demonstrate the importance and presence of isomiRs[42]. To address the age specific expression of isomiRs, single nucleotide resolution is required. Improved library preparation and sequencing methods together

with increasing read numbers per sample will likely allow for an in-depth characterization of isomiRs in challenging specimens such as whole blood.

Another aspect for respective studies is the underlying specimen type. A literature search reveals that for human miRNA biomarker studies mostly plasma, serum, and blood cells (either PBMCs or whole blood) are considered with a more recent trend towards exosomes. Since we are interested in the connection of miRNA expression and the immune system by analyzing multiple diseases[43] we measured blood cells. Different aspects can be used to provide an even more comprehensive systemic picture of miRNAs and aging. First, the cell free part of the blood is also correlated to miRNA aging[44,45]. One important aspect are vesicles. Cellular senescence for example contributes to age-dependent changes in circulating extracellular vesicle cargo[46]. Moreover, the differential loading of vesicles is correlated to different human diseases[47–49]. Likewise, for the cellular part, resolution can be increased. For example, the miRNomes could be investigated per blood cell type[50]. One challenge is in that the purification of the different cell types by different isolation techniques potentially alters the miRNA content. Positive and negative selection, as well as Fluorescence-activated cell sorting (FACS) have a highly significant influence on the physiological miRNA content[32]. Here, single cell miRNA profiling might help to improve our understanding of age-related miRNA patterns in the future. At best, single cell miRNA data and cell free miRNA profiles are combined in the future using advancing sequencing technologies. Finally, such data might further our understanding of miRNAs in aging, diseases and their interplay with organ patterns that are only partially understood[29,51].

Over recent years, numerous studies have emerged highlighting systemic molecular aging factors detected with different omics technologies, including epigenetics, transcriptomics, and proteomics. Our study specifically extends our knowledge of blood and plasma-based miRNA patterns in aging. In our study we observe non-linear miRNA aging patterns. Moreover, the high degree of age-related biomarker patterns challenges the concept of age independent miRNA biomarker profiles, calling for different statistical models in aged and younger individuals. The changes with aging are not only attributed to one mature form, we also provide detailed insights into changes of the usage of the 3' and 5' mature arms in aging.

Furthering our understanding of age-related miRNA changes in healthy individuals and diseased patients will not only increase our understanding of age-related blood-borne gene regulation, but also improve miRNA-based biomarker development, and aid the development of RNA-based therapies.

## Methods

**Cohort**. In this study, we processed data from $n_{total} = 4433$ whole blood samples. We excluded 40 individuals (0.9%) because of insufficient data quality or missing clinical or demographic information. The final cohort consists thus of 4393 samples. These include unaffected controls ($n_{HC} = 1{,}334$), Parkinson's Disease ($n_{PD} = 944$), heart diseases ($n_{HD} = 607$), non-tumor lung diseases ($n_{NTLD} = 586$), lung cancer ($n_{LC} = 517$), and other diseases ($n_{OD} = 405$). The diseases can be split further in sub-classes. For lung cancer, we included non-small cell, as well as small

cell lung cancer. For non-small cell lung cancer, we can further divide them in adenocarcinoma and squamous cell carcinoma. These split in low grade and high-grade tumors according to the TNM grading. The lung cancer cohort has been previously described in more detail[52]. The heart diseases include coronary artery disease, dilated cardiomyopathies and acute coronary syndrome. The non-tumor lung diseases include mostly chronic obstructive pulmonary diseases, the other diseases include sepsis, liver cirrhosis, breast cancer, endometriosis, and melanoma patients. We aggregate the diseases to an organ level (heart, brain and lung). Only for the lung we split the cohort in cancer and non-cancer samples. This aggregation level has been selected in a manner to be able to distinguish between healthy and diseased aging by having sufficient cohort sizes. Detailed diagnoses for each sample are provided in Supplementary Data 1. All participants gave informed consent. The local ethics committee of Saarland University approved the study. The study has been conducted in compliance with all relevant ethical regulations regarding the use of human study participants.

**RNA extraction and measurement of miRNAs.** RNA from 4433 whole blood samples in PAXgeneTubes (BD Biosciences, Franklin Lakes, NJ, USA) was isolated using the PAXgene Blood miRNA Kit (Qiagen, Hilden, Germany) using manufacturers recommendation. The extractions were done manually or semi-automatically on the Qiacube robot. The RNA was quantified using Nanodrop (Thermo Fisher Scientific, Waltham, MA, USA) and the RNA integrity was checked using a bioanalyzer with the RNA Nano Kit (Agilent Technologies, Santa Clara, CA, USA). The genome-wide miRNA expression profiles of human mature miRNAs was determined with Human miRNA microarrays and the miRNA Complete Labeling and Hyb Kit (Agilent Technologies). The labeled RNA was hybridized to the arrays for 20 h at 55 °C with 20 rpm rotation. The microarrays were subsequently washed twice, dried and scanned with 3 μm resolution in double-path mode (Agilent Technologies). The raw data were extracted using the manufacturers Feature Extraction software (Agilent Technologies). Details on the RNA extraction and microarray measurement procedure have been also previously described[53,54]. In difference to our previous studies we tried to further minimize any variability. In this study, we thus only included genome wide miRNA profiles that have been measured using the Agilent miRBase V21 biochip.

**Blood cell deconvolution.** To analyze the miRNA blood cell composition, we made use of our previous study that presented a high-resolution representation of human miRNAs in different blood compounds[50]. From the data, we asked which miRNAs are present in at least one sample of the respective blood compound and generated an upset plot from the data. In some detail, we included serum, microvesicles, red blood cells, CD15, CD19, CD8, CD56, CD4, and CD14 cells.

**Correlation of age and sex to miRNAs.** To find associations between the sex and the miRNA expression we applied 2-tailed non-parametric Wilcoxon Mann–Whitney tests. To compute linear correlation values between the age and miRNA expression values we computed the Pearson Correlation Coefficient (PC) and Spearman Correlation (SC). Further, to detect potentially non-linear relations between single miRNAs and the age we also computed the Distance Correlation (DI) between age and sex. To relate the DI and the SC, we computed a smoothed spline with eight degrees of freedom and computed the minimal Euclidean distance of each data point from the spline. Points with a distance of 0.02 (the threshold of 0.02 has been computed by a histogram-based approach) were highlighted and are considered to follow a non-linear trend with aging. In the further analyses, we applied only the rank-based Spearman Correlation (SC) instead of the Pearson Correlation that assumes linear effects in data. Beyond linear and non-linear correlations between single miRNAs and the age we also performed different standard dimension reduction technologies, including principal component analysis, t-stochastic neighborhood embedding (t-SNE) and Uniform Manifold Approximation and Projection (UMAP). To calculate the fraction of variance attributed to the age and sex we applied principal variant component analysis (PVCA), originally developed to discover batch effects in microarray experiments.

**Analysis of arm shift events.** Recently, we developed the miRSwitch database and analysis tool to identify and characterize human arm shift and arm switch events[30]. To detect associations between aging and differential arm usage we considered the following criteria. First, the percentage of the 5' mature arm given the total expression of 3' and 5' arm must correlate with an absolute Spearman Correlation Coefficient > 0.2. Second, the correlation must reach a p-value of at least 0.05. The p-value is computed by the R cor.test function via the asymptotic t approximation. Third, the difference between the minimal and maximal percentage of 5' arm expression for any samples must exceed 0.2 (20%). As fourth and last condition, the 3' and 5' mature form must have a different sign, i.e., the 5' has to increase with age and the 3' to decrease or vice versa. The miRNAs that were discovered by this procedure where then checked by miRSwitch.

**Cluster analysis and miRNA enrichment analysis.** We split the miRNAs in 5 groups, strongly decreasing with age (SC < −0.2), decreasing with age (SC between −0.2 and −0.1), not changing with age (SC between −0.1 and 0.1), miRNAs increasing with age (SC > 0.1 and <0.2) and miRNAs increasing strongly with age

(SC > 0.2). For each cluster, we computed smoothed splines for each miRNA and the cluster average allowing three degrees of freedom. Further, we computed for disjoint age windows of five years whether miRNAs are significantly higher or lower in cases versus controls at an alpha level of 0.05 and colored them, respectively, in red and green. To find categories that are significantly enriched either for miRNAs increasing or decreasing over age we performed a miRNA enrichment analysis using the miEAA tool[55], which has been recently updated[56]. Thereby, for over 14,000 categories running sum statistics are computed. The sorted list of miRNAs (increasing correlation with age) is processed from left to right. Whenever a miRNA is located in a category the running sum is increased otherwise it is decreased. The running sum is then plotted along with 100 random permutation tests. Notably, the p-value is not computed from the permutations but exactly by using dynamic programming. A category showing a perfect "V" like shape would contain miRNAs that are increasing over age while a category following a pyramid like shape contains miRNAs that are decreasing over age.

**Sliding window analysis based on Cohen's d.** Since p-values rely on the effect size and the cohort size different group sizes bias the results frequently. In our sliding window analysis, we observed substantial differences, i.e., cases and controls are not equally distributed across the age range. We thus performed all analyses using Cohen's d as effect size. All effects with an absolute value of above 0.5 were considered relevant. Negative effect sizes thereby characterize downregulation and positive effect sizes upregulation. We computed effect sizes for each disease in windows of 10 years, shifted by one year, starting from 30 and ending at 70 years (i.e., the last window is from 70 to 79 years). Only when at least 20 cases and control measurements were available effect sizes were computed. The calculated effect sizes were then summarized and a smoothed spline with eight degrees of freedom were computed.

**Self-organizing map (SOM) for finding disease patterns.** One task in high dimensional data analysis is to group features and to generate lower dimensional representation of high dimensional data. Self-organizing maps (SOMs) are one type of artificial neural networks (ANNs), relying on competitive learning. As described by Kohonen already in 1982[57], in a network of adaptive elements "receiving signals from a primary event space, the signal representations are automatically mapped onto a set of output responses in such a way that the responses acquire the same topological order as that of the primary events". From input data, a typically two-dimensional discretized representation of the input space is derived that can be visualized by heat maps. To compute self-organizing maps for patients and controls in an age dependent manner we computed the effect size for each disease group over all patients, for young patient (30–60 years) and for old patients (60–80 years) separately. Only 801 highest expressed miRNAs were included in this analysis. For the biomarker sets, a $10 \times 10$ hexagonal som grid was used to train a network. The data set was presented 10,000 times to the network. The learning rate was set to be between 0.05 and 0.01, meaning that the learning rate linearly decreased from 0.05 to 0.01 over the 10,000 iterations. To cluster the SOM cells, we performed hierarchical clustering. In more detail, we applied the R hclust function to carry out agglomerative complete linkage clustering. As distance measure we computed the Euclidean distance using the R dist function.

**Plasma proteomics measurements.** We used data from a recent study investigating the effect of aging on the human plasma proteome. In this study, 2925 proteins were measured using the SomaScan assay in 4264 subjects from the INTERVAL and LonGenity cohorts[5]. The SomaScan platform is based on modified single-stranded DNA aptamers binding to specific protein targets. Assay details were previously described. Relative Fluorescence Units (RFUs) were log10-transformed and we used a 10 years sliding window to estimate proteins trajectories throughout lifespan.

**Target analysis and target network analysis.** The main biological function of miRNAs is to bind the 3' UTR of genes and to degrade the target mRNAs. In reality, miRNAs and genes thereby follow a n:m relation, i.e., one miRNA can regulate many genes and one gene is regulated by many miRNAs. Further, there exist different confidence levels to assume a pair-wise regulation of a miRNA to a target gene. Most relations are only predicted by one or several computational analyses. Another set is composed of miRNA gene pairs with weak evidence, e.g., from microarray experiments. The most reliable category consists of miRNA gene pairs with strong evidence, e.g., validated by reporter assays. We only considered this most reliable set of miRNA gene interactions and extracted the set from the miRTarBase database[34,58]. Our analysis highlighted that around 20% of miRNAs are increasing with age, 20% are decreasing and 60% are not age dependent. We assumed the same distribution for human plasma proteins changing with age and asked how many miRNAs going down with age regulate genes/proteins going up and down with age, respectively. Similarly, we asked how many miRNAs going up with age regulate genes/proteins going up and down with the age.

To construct a reliable core network, we combined five stringed filtering approaches and only considered those connections between miRNAs and genes that fulfill all filtering criteria. In the least stringent version the filters include (a) a strong experimental evidence of a target interaction from the literature; (b) one of

the most decreasing miRNAs (5%) regulates (c) one of the most upregulated proteins (5%) over aging. To avoid a bias towards genes/proteins that are targeted only by one or few miRNAs, potentially also fragmenting the network, we (d) only considered proteins that are regulated by more than eight miRNAs. Next, we analyzed the correlation between miRNAs and genes/proteins in the network over 40 discrete age ranges from 30 to 70 years. Each age range thereby spans 10 years. For the 40 data points corresponding to 40 age windows we computed the Spearman correlation between miRNA expression in this age window and protein expression. As last criterion we added (e) only edges that have an absolute Spearman correlation of at least 0.6. This network has been visualized with the igraph library. Nodes were colored with respect to changes in age and edges weights relative to the absolute Spearman correlation.

**Single cell analysis**. We used data that have been made available by 10× genomics (https://support.10xgenomics.com/single-cell-gene-expression/datasets/3.0.0/pbmc_10k_v3). The profiles were subsequently processed with scater[59] and scran[60] with default parameters, cell type annotations with singleR[61].

**Reporting summary**. Further information on research design is available in the Nature Research Reporting Summary linked to this article.

## Data availability
The raw microarray measurements are freely available for any scientific purpose upon request as Excel Table and Tab Delimited Text file (110 MB) to data@ccb.uni-saarland.de. The use of the data for commercial purposes is prohibited.

## Code availability
The data analysis has been performed using the R software for statistical computation (R 3.3.2 GUI 1.68 Mavericks build (7288)) using freely available packages. The following packages were used: ROC, RColorBrewer, preprocessCore, tsne, effsize, UpSetR, kohonen, fmsb, igraph. All packages are available from Bioconductor or CRAN.

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

## Acknowledgements

This work was supported by grants from the Michael J Fox Foundation (to Andreas Keller) and the Luxembourg National Research Fund (FNR) within the PEARL programme (FNR; FNR/P13/6682797 to Rejko Krüger).

## Author contributions

T.F.: Data analysis, conception of the study and analyses; B.L.: Data analysis, manuscript drafting; N.S.: Data interpretation, manuscript drafting; O.H.: Data interpretation, manuscript drafting, data representation; M.K.: Data analysis; Y.L.: Data interpretation; N.G.: Data interpretation, data representation; L.G.: Data interpretation; C.B.: Data analysis; R.B.: Data interpretation, conception of the study and analyses; F.K.: Data analysis, data representation; R.K.: Data interpretation, conception of the study and analyses, providing clinical data and patient specimens; F.L.: Data interpretation, providing clinical data and patient specimens; N.L.: Performing analyses and contributing experimental data; B.M.: Data interpretation, conception of the study and analyses, providing clinical data and patient specimens; B.F.: Data interpretation, manuscript drafting; W.M.: Data interpretation; D.B.: Data interpretation; K.B.: Data interpretation; C.D.: Data interpretation; A.K.v.T.: Data interpretation, providing clinical data and patient specimens; G.W.E.: Data interpretation, providing clinical data and patient specimens; S.M.: Data interpretation, Performing analyses and contributing experimental data; N.B.: Data interpretation, Performing analyses and contributing experimental data; M.R.: Data interpretation, providing clinical data and patient specimens; T.W.C.: Data interpretation, manuscript drafting; E.M.: Data interpretation, conception of the study and analyses, manuscript drafting; A.K.: Data analysis, Data interpretation, conception of the study and analyses, manuscript drafting.

## Funding

## Competing interests

M.K. is also employed by Hummingbird Diagnostic GmbH. The remaining authors declare no competing interests.
