## [Peer Review File · Nature Communications]

REVIEWER COMMENTS

Reviewer #1 (Remarks to the Author):

In this study by Fehlmann et al., the authors performed a large-scale blood-borne miRNA profiling among both healthy (n=1334) and diseased individuals (n=3059), covering a wide and high resolution of age range from 40-80. The selection of statistical and bioinformatics approaches was considerable and appropriate. This study has a high impact as it adds new knowledge on 1) potential non-linear patterns of miRNA expression with age; 2) “critical window” throughout a lifetime that miRNA expressions change dramatically by different major diseases; 3) the strong intrinsic miRNA expression pattern with aging despite the change of blood cell subpopulation. These messages are important for future miRNA study profiled in whole blood. Though the clarity of the results and methods needs to be improved, I only have several minor comments:

1. Line 106: “While 231 miRNAs 106 overlapped between these groups, this number was not significant ($p=0.18$).” What statistical test was used here?
2. Lines 110-113. It is unclear what is the purpose of building a miRNA age. Related results should be presented somewhere as a figure (e.g., a scatterplot between chronological age vs. miRNA age) and tables (a list of miRNAs selected by LASSO).
3. Fig. 2 legend: “represent examples provided in Fig. 2E”- should be Fig. 2G.
4. Fig. 2- The green color coding for group “4” coincides with the “increasing with age”- consider another distinguishable color.
5. I think in general, Fig. 2 would benefit from a little more explanatory text on the actual images themselves- there’s a lot of clustering and color-coding going on, and it’s hard for me to tell if these are meant to be the same across different sub-figures or different. I would also consider removing or changing Fig 2H- the color-coded body adds to the confusion, and in my opinion is a little too crowded to be all that informative. I think that space would be put to better use explaining the rest of the figure a little more. A few graphic legends to provide reference to the color-coding without having to flip back and hunt through the main figure legend would go a long way.
6. Fig 3E legend: “The miRNA is significant up-regulated in lung cancer”- should be down-regulated?
7. Fig 3F legend: “down-regulated in heart diseases of younger patients”- it is confusing how the authors concluded this? The green line is roughly around effect size = 0. Same for PD and LD- it is unclear how the figure supports the conclusions.
8. The color-coding for lung cancer and parkinson’s in Figs. e-f is hard to distinguish.
9. Cite miEAA 2.0, if applicable.
10. Please add a brief introduction of SOM (what it is and what it does) in the method or when it is first mentioned. As the authors presented the grid, it would be helpful to annotate the hexagonal

grid to make it more informative (e.g., what are the functions of those clustered down- or up-regulated miRNA grids).

11. Perhaps the authors were using default settings, but please explicitly state the distance and linkage methods used in the hierarchical clustering.

12. Fig. 4A and 4B were not mentioned in the results except in the legend- what is the purpose of presenting 4B? The title of 4B is "HC vs. DIS" (which were not defined anywhere), but I guess it means healthy control vs. diseased rather than young vs. old?

13. The methods for the regulatory network analysis were missing.

14. Any explanations of why SEMA3E was not detectable, given it was one of the hub proteins in the core network?

Reviewer #2 (Remarks to the Author):

This study analyzed changes in age-related microRNAs (miRNAs) by analyzing whole blood from 1334 individuals. In addition, miRNA profiles of 2059 patients with disease were analyzed to investigate alterations in miRNA profiles. Investigations of blood miRNA profiles depending on sex and age found that a greater number of miRNAs were found to be correlated with age (1568 for age and 362 for sex). Vast majority of the presented data were based on in silico analysis and were not so impressive unfortunately because experimental validations were lacking. My specific comments follow:

1. The most critical issue is why the authors analyzed whole blood miRNAs but not plasma or serum miRNAs. Indeed, extracellular circulating miRNAs are potential biomarkers for various diseases and the mechanisms of trafficking disease-related (or age-related) miRNAs to extracellular vesicles would be more intriguing.

2. For the analysis in fig 5, the authors used their previous data of blood cell composition. But ideally, whole blood miRNA and each blood cell miRNA pairs should be analyzed in the same participants. The results of the association between whole blood miRNAs and age strongly suggest that miRNA profiles in each blood cell type would be associated with age.

3. Fig 5c: it suggests the distribution of serum miRNAs and exosomal miRNAs seems largely different, which is a bit weird result. How do you explain it?

4. Fig 6g: the labels of axis are shown as "t-SNE" but the legend describes as "UMAP".

Reviewer #3 (Remarks to the Author):

Aging is a key risk factor for chronic diseases of the elderly and has been studied in several global contexts, including epigenetic factors, proteins, metabolites, and RNAs including microRNAs. The authors identified previously unobserved nonlinear changes in age-related microRNAs by analyzing whole blood from 1,334 healthy individuals. This study provides a foundation for understanding the relationship between healthy aging and disease, and for the development of age-specific disease biomarkers. Generally, this study may be potential interesting for the relevant study, especially for the effect of age in diverse human diseases, but temporal expression patterns of miRNAs should be analyzed in this study, and in-depth analysis based on the multiple isomiRs should be mentioned. More specific comments are listed below:

Major comments:

1. Generally, references should be not cited in abstract section. In the current abstract, there were 2 references.
2. For miRNA profiles, authors found that “miRNA profiles are strongly associated with age but independent of sex”, but some studies have shown that some miRNAs may be diverged between males and females. It is better to present these difference based on the different studies.
3. For miRNAs in miRBase database (Version 21.0), why not select the novel version (22.0 or 22.1)? Further, the phenomenon of multiple isomiRs have been widely detected in multiple animal species, especially in diverse human diseases, it is better to show the detailed isomiR landscapes based on the miRNA profiles. As you know, each miRNA locus always yields several dominant isomiRs, and some isomiRs may be involved in diverse sequences or even shifted seeds. I think the systematic analysis based on isomiRs will provide more information for this study.
4. More important, miRNA profiles were analyzed based on blood samples from diverse human diseases, including heart disease and lung diseases. However, some of the small RNAs were diverged expressed in diverse tissues. It is quite important to show the temporal expression patterns for these miRNAs.
5. For figure legends, it is better to show the total description.
6. Finally, the statistical analysis should be performed for the data analysis. Many results (such as Figure 3A, Figure 3b, et al.) should present the detailed p values to show whether it exists significant difference between or among groups.

Reviewer #1 (Remarks to the Author):

In this study by Fehlmann et al., the authors performed a large-scale blood-borne miRNA profiling among both healthy (n=1334) and diseased individuals (n=3059), covering a wide and high resolution of age range from 40-80. The selection of statistical and bioinformatics approaches was considerable and appropriate. This study has a high impact as it adds new knowledge on 1) potential non-linear patterns of miRNA expression with age; 2) “critical window” throughout a lifetime that miRNA expressions change dramatically by different major diseases; 3) the strong intrinsic miRNA expression pattern with aging despite the change of blood cell subpopulation. These messages are important for future miRNA study profiled in whole blood. Though the clarity of the results and methods needs to be improved, I only have several minor comments:

1. Line 106: “While 231 miRNAs 106 overlapped between these groups, this number was not significant (p=0.18).” What statistical test was used here?

The p-value has been computed by Fisher’s exact test. While we originally computed and reported a one-sided test, we now use a two-sided test bringing the p-value to 0.35. Additionally, we performed a Pearson’s Chi-squared Test yielding a concordant p-value of 0.36. We added this information in the revised manuscript.

2. Lines 110-113. It is unclear what is the purpose of building a miRNA age. Related results should be presented somewhere as a figure (e.g., a scatterplot between chronological age vs. miRNA age) and tables (a list of miRNAs selected by LASSO).

Respective regression models have already been proposed for several other omics entities (such as epigenetics and proteomics studies). An example is given in a recent Nature Medicine article entitled “Undulating changes in human plasma proteome profiles across the lifespan” published by the Wyss-Coray lab. We have been interested to see whether we can build a respective model for miRNAs and whether such a model performs better or worse than other omics types. Indeed, the performance was marginally below the one for the proteins. Since the cohorts of these studies and our study are not comparable making it hard to judge whether the differences come from the cohort or from the actual miRNA signature and since this analysis represents only a minor aspect of our work we decided to remove it, instead of making it more prominent by additional graphics and tables.

3. Fig. 2 legend: “represent examples provided in Fig. 2E”- should be Fig. 2G.

Thanks for pointing us at the error. We corrected this wrong label in the revised manuscript.

4. Fig. 2- The green color coding for group “4” coincides with the “increasing with age”- consider another distinguishable color.

We tried to use less and in general more consistent colors (please see also our response to your comment below). In fact, there is no need to color the four numbers since the numbers already contain the relevant information. Thus, we kept the red and green color for increasing and decreasing with age, respectively, and removed the color from the numbers in Fig. 2F. In Fig. 2G we left the colors since other color schemes caused an issue with the semi-transparency.

5. I think in general, Fig. 2 would benefit from a little more explanatory text on the actual images themselves- there’s a lot of clustering and color-coding going on, and it’s hard for me to tell if these are meant to be the same across different sub-figures or different. I would also consider removing or changing Fig 2H- the color-coded body adds to the confusion, and in my opinion is a little too crowded to be all that informative. I think that space would be put to better use explaining the rest of the figure a little more. A few graphic legends to provide reference to the color-coding without having to flip back and hunt through the main figure legend would go a long way.

We understand that we used too many different colors in the figure, partially even inconsistent between the sub-panels. We modified the figure as follows:

- *Panel b: The figures from the boxes were removed since they were redundant with the labels for sex and age.*
- *Panel c: We removed the arrows and labels within that panel. Instead, we now provide a legend in the lower right part of the image.*
- *Panel f: We removed the color from the circles identifying the focus areas in panel g. The information given by these colors were redundant with the numbers.*
- *Panel h: The intention of the panel was to create a visualization that highlights that basically all organs are affected. Since a graphical representation adds to this message we decided to leave the body map. However, we fully agree with the reviewer that the colors are misleading. We now show all organs that are significant in grey color. Furthermore, we improved the table in this panel.*

6. Fig 3E legend: “The miRNA is significant up-regulated in lung cancer”- should be down-regulated?

Actually, the example is intended to show an up-regulated miRNA in lung cancer. We however mixed the labels between Fig. 3E and Fig. 3F. The example of the up-regulated miRNA in lung cancer has been by fault shown in panel F and vice versa the unspecific miRNA in panel E. This error has been corrected in the revised manuscript. We apologize for this confusion. Please see also the comment to your point 7.

7. Fig 3F legend: “down-regulated in heart diseases of younger patients”- it is confusing how the authors concluded this? The green line is roughly around effect size = 0. Same for PD and LD- it is unclear how the figure supports the conclusions.

As mentioned in our reply to your previous point 6 we mixed the legends of Figure 3E and 3F. This has been corrected in the revised manuscript. Again, we apologize for this inconvenience.

8. The color-coding for lung cancer and parkinson’s in Figs. e-f is hard to distinguish.

We agree that the color code using the two different shades of blue was hard to distinguish. We now show lung cancer in grey. Similar to the comment 5, we also avoided using colors that are not required. The green circles have been modified to grey color.

9. Cite miEAA 2.0, if applicable.

We appreciate this comment. When we submitted the original version of this manuscript miEAA2 was not accepted for publication. In the meantime, the version 2 has been published and we provide a reference to the most recent version.

10. Please add a brief introduction of SOM (what it is and what it does) in the method or when it is first mentioned. As the authors presented the grid, it would be helpful to annotate the hexagonal grid to make it more informative (e.g., what are the functions of those clustered down- or up-regulated miRNA grids).

SOMs play a relevant role in our study and are actually less common than expected. We appreciate the suggestion to add information on SOMs. We added not only an introduction to SOMs in the Methods part but also motivated why we use them for our results. Please see in this context also our reply to your comment 12.

In the original manuscript we missed to add the annotation of the 10x10 grid. We now provide this annotation in the revised manuscript. This includes information on the distribution of miRNAs and also a detailed supplemental table that shows, which miRNAs cluster together. Following your

suggestion, we also performed a miRNA set enrichment analysis using miEAA2 for the 100 cells, as we did for the other miRNA sets. In this analysis, we see one challenge: As we describe in the revised results section, the miRNAs per cell are not equally distributed but rather follow a normal distribution with a mean value of 5 miRNAs per cell and a small second peak at 17 miRNAs per cell. We have shown this for your convenience as histogram on the right side. These larger cells are located in the center of the SOM grid and are enriched for the miRNAs not being differentially expressed. In contrast, the cells with significantly deregulated miRNAs are typically smaller leading to artificial pathway enrichment results. The obvious solution was to run the analysis with fewer cells. By doing so, the enrichment analysis became, however, similar to the standard analyses of “up- versus “down-regulated” miRNAs. We now mention the miRNA distribution per cell and strengthen the message that both, the general patterns of the clustering and the miRNAs per cell are informative while a pathway analysis per single SOM cell might be misleading. Finally, the annotation of the SOM grid is provided in the new Supplemental Table 6.

11. Perhaps the authors were using default settings, but please explicitly state the distance and linkage methods used in the hierarchical clustering.

We used agglomerative complete linkage clustering. As distance metric we computed the Euclidean distance. We added this information along with the used packages to the Material section of the revised manuscript.

12. Fig. 4A and 4B were not mentioned in the results except in the legend- what is the purpose of presenting 4B? The title of 4B is “HC vs. DIS” (which were not defined anywhere), but I guess it means healthy control vs. diseased rather than young vs. old?

As also indicated in the response of your comment 10 we failed to emphasize the importance of the SOM analysis and to motivate why we used it. In fact, we only briefly mentioned the Figures 4A-C in the original manuscript and missed to indicate why we placed these figures in this context. We now substantially improved this part of the results section. We performed the analysis to demonstrate that aging in general has a more substantial influence as compared to diseases in general, as indicated by the colors in the SOM grid. This might, however, be misleading since effects of different diseases could cancel each other out, e.g. a miRNA that is upregulated in lung cancer, but not deregulated in heart diseases and down-regulated in PD will on average be not deregulated in the global comparison of controls versus diseases. This led us to a detailed per-disease comparison. This part has also been substantially improved in the results section of the revised manuscript. As you speculate, “HC vs DIS” is healthy control versus diseases. We of course explain this in the revised manuscript.

13. The methods for the regulatory network analysis were missing.

As you indicated we did not provide sufficient information of the network analysis, which was described only in one single sentence in the original manuscript. We now provide a much more detailed description that facilitates to reproduce the results.

14. Any explanations of why SEMA3E was not detectable, given it was one of the hub proteins in the core network?

Based on our present data and data from other groups, we can only speculate about the reason. The first and most obvious explanation is to claim an issue of single cell resolution. To test this hypothesis, we checked data from the GTEX consortium that provide evidence that the gene is not expressed in bulk data on whole blood level. Since the according protein is, however, expressed in plasma, one might speculate about an indirect mode of regulation, e.g. via exosomes. We kindly ask you to consider also our reply to reviewer 2, point 1. Sequencing of exosomes from blood is performed in current aging experiments and we will use these data to test this hypothesis in the future.

Reviewer #2 (Remarks to the Author):

This study analyzed changes in age-related microRNAs (miRNAs) by analyzing whole blood from 1334 individuals. In addition, miRNA profiles of 2059 patients with disease were analyzed to investigate alterations in miRNA profiles. Investigations of blood miRNA profiles depending on sex and age found that a greater number of miRNAs were found to be correlated with age (1568 for age and 362 for sex). Vast majority of the presented data were based on in silico analysis and were not so impressive unfortunately because experimental validations were lacking. My specific comments follow:

The data set that is the basis of our work contains over 200,000,000 (2×10^8) single miRNA measurements. It is evident that the results presented on such a high dimensional data set calls for substantial in-silico analyses. This holds even more true with the annotation data being also multi-dimensional (diseases \times time-points). We consider this strong computational component, including classical biostatistics, bioinformatics and also deep learning and neural network aspects as one strength of our study. We stress this aspect in the revision of the manuscript.

We do not agree with the assessment of the value of our data. As for the findings of age and sex dependent miRNAs we validated existing knowledge on an impressive data set. In addition, our work presents many novel aspects and also raises new hypotheses. While non-linear aging component are known for proteomics data¹, this aspect has never been evaluated at scale for small non-coding RNAs. In the revised manuscript we also demonstrate an age-related shift to 5' mature expression. The most important novel finding is, however, the extent to which aging blurs biomarker patterns. The dependency of physiological aging and miRNA expression is lost to a significant degree. These results challenge current biomarker studies and call for statistical models that rely on age-dependent biomarkers. In the revised manuscript we highlight both the novelty of our central findings and their implications. Of course, we agree that further targeted validation might strengthen the manuscript even more, but as explained here we are convinced that the results of our work are of general interest for the research community and of particular importance for the miRNA and/or aging community.

1. The most critical issue is why the authors analyzed whole blood miRNAs but not plasma or serum miRNAs. Indeed, extracellular circulating miRNAs are potential biomarkers for various diseases and the mechanisms of trafficking disease-related (or age-related) miRNAs to extracellular vesicles would be more intriguing.

We agree that plasma and serum miRNAs are excellent biomarkers for many human diseases. Moreover, we agree that vesicles are important carriers of miRNAs. However, whole blood has likewise an immense potential as source for human biomarkers including RNA. This is not only demonstrated by our own research on cellular blood miRNA signatures (in ~50 manuscripts; some selected include²⁻⁹) but also substantiated by the literature with hundreds of

manuscripts. From our perspective the choice of matrix (blood cells, serum , ...) depends on the biological hypothesis to be tested. We summarized the advantages and disadvantages of the different matrices such as blood- and plasma-based miRNAs in several reviews (e.g. ^{10,11}).

In the present study, we decided for several reasons to measure whole blood and not plasma or serum.

First, PAXGene tubes use whole blood and lyse cells immediately upon blood drawing. This allows to largely freeze the status of the RNA and as a result to insure a high degree of standardization for the collection of the samples. This is especially important for the analysis of an extended number of samples. By contrast, the protocols for the isolation of specific blood components like plasma require different steps for sample processing and are as result far more prone to variations. This is problem is even more evident for the isolation of extracellular vesicles, the purification and characterization of which are far more demanding than frequently suggested.

Second, our previous studies show that the majority of RNA isolated from whole blood stems from lysed blood cells. This facilitates to trace specific RNAs to their cells of origin in future studies. By contrast, it is near to impossible to trace the origin of RNAs isolated form plasma or extracellular vesicles.

Third, our research on human diseases and our more recent aging studies ^{12,13} point at an important role of the immune cells on the RNA pattern. We published a substantial number on PAXGene biomarker patterns that have partially been highly cited and validated by others, providing evidence that this approach is well suited to measure age, disease and immune cell related miRNA patterns. One of the most comprehensive evaluations of the experimental system was published already in 2011 in Nature Methods ¹⁴. The results of the present study suggest that the carriers of the aging information in the blood are indeed the white blood cells (see for example Figure 5B, indicating that 75% of the signal come from white blood cells, 5 from red blood cells and 25 from the cell free content). In the revised manuscript, we provide a justification and explanation on why we measured whole blood signatures.

2. For the analysis in fig 5, the authors used their previous data of blood cell composition. But ideally, whole blood miRNA and each blood cell miRNA pairs should be analyzed in the same participants. The results of the association between whole blood miRNAs and age strongly suggest that miRNA profiles in each blood cell type would be associated with age.

We fully agree with the reviewer that the different blood cell types will most likely contain miRNAs that are associated with age. We explored miRNA patterns in different blood cell types in several previous studies. Measuring such patterns per blood cell type bears however substantial experimental challenges. The main challenge is to measure signals that closely reflect the physiological situation. This is not easy to accomplish with samples that stem from different clinics. Either the different blood cell types are isolated in each clinic after blood drawing or the samples will be shipped to a central unit where the cell separation takes place. For the first option there is the problem of comparability of independent isolation and for the second option, the problem of storage and shipping conditions. We previously showed that the cell separation and purification have an immense impact on the cellular miRNA patterns. One example from a previous study is presented on the right ¹⁵. Depending on whether positive selection, negative selection or FACS sorting is performed the miRNA content varied substantially. Many miRNAs are for example only present in samples that were obtained by positive selection or FACS. Since there is a substantial overlap with our aging miRNAs, adding cellular patterns might rather blur the aging patterns. Finally, we would also like to mention a challenge, which is related to workload and costs. In the study, we included 4,500 individuals. If we would measure the most important 6-8 cell types this would mean 27,000 to 36,000 complete miRNomes, which requires several years of measurement work and costs of 4-8 million Euro.

One example from a previous study is presented on the right ¹⁵. Depending on whether positive selection, negative selection or FACS sorting is performed the miRNA content varied substantially. Many miRNAs are for example only present in samples that were obtained by positive selection or FACS. Since there is a substantial overlap with our aging miRNAs, adding cellular patterns might rather blur the aging patterns.

Finally, we would also like to mention a challenge, which is related to workload and costs. In the study, we included 4,500 individuals. If we would measure the most important 6-8 cell types this would mean 27,000 to 36,000 complete miRNomes, which requires several years of measurement work and costs of 4-8 million Euro.

To address the abovementioned challenges, which we now discuss in the manuscript, we would like to propose single cell miRNA sequencing as future solution. While single cell sequencing has become standard for RNA analyses it is still under development for miRNA research. We recently tested 18 protocols and variations and are convinced that the according experimental technologies will rapidly improve. We also added information on this issue to the discussion of the revised manuscript.

3. Fig 5c: it suggests the distribution of serum miRNAs and exosomal miRNAs seems largely different, which is a bit weird result. How do you explain it?

As you stated, miRNAs in serum and in exosomes are shared generally to a large degree. A comprehensive study by Cheng and co-workers provides deep insights into the distribution of miRNAs in PAXGene blood, serum, plasma and vesicles¹⁶. We share a figure from the work by Cheng on the right. As you can see, miRNAs are partially specific for plasma/serum or exosomes. These data match our results largely, as e.g. shown in Figure 5A.

To specifically address your question regarding Figure 5C we want to clarify that the Figure does not show a largely different distribution of serum miRNAs and exosome miRNAs. In this Figure we split all miRNAs to the degree of variability with age.

The figure suggests that miRNAs increasing or decreasing with age are not randomly distributed in serum and exosomes but show enrichment in the one or the other fraction. This is in-line with many studies providing evidence of exosomes as specific transporters of miRNAs. We clarified this point in the revised manuscript.

4. Fig 6g: the labels of axis are shown as “t-SNE” but the legend describes as “UMAP”.

We apologize this error. We corrected the labeling of the Figure. The single cell data dimension reduction for this study has been computed using UMAP.

Reviewer #3 (Remarks to the Author):

Aging is a key risk factor for chronic diseases of the elderly and has been studied in several global contexts, including epigenetic factors, proteins, metabolites, and RNAs including microRNAs. The authors identified previously unobserved nonlinear changes in age-related microRNAs by analyzing whole blood from 1,334 healthy individuals. This study provides a foundation for understanding the relationship between healthy aging and disease, and for the development of age-specific disease biomarkers. Generally, this study may be potential interesting for the relevant study, especially for the effect of age in diverse human diseases, but temporal expression patterns of miRNAs should be analyzed in this study, and in-depth analysis based on the multiple isomiRs should be mentioned. More specific comments are listed below:

We appreciate your generally positive and encouraging comments. The analysis of the temporal miRNA effects was one of the most relevant aspects of our study and we further strengthened this aspect. While an analysis of isomiRs is not feasible due to the underlying experimental technology (please see also our reply to your point 3 for details) we understand that we did not leveraged the full potential of the data set. Besides isomiR analysis, arm shifting is increasingly becoming an important focus of miRNA research. The experimental technique is very well suited to analyze respective arm shifts and arm switches and we added according results to the manuscript.

Major comments:

1. Generally, references should be not cited in abstract section. In the current abstract, there were 2 references.

We removed the references from the abstract of the revised manuscript and payed attention to cite the relevant literature in the Introduction.

2. For miRNA profiles, authors found that “miRNA profiles are strongly associated with age but independent of sex”, but some studies have shown that some miRNAs may be diverged between males and females. It is better to present these difference based on the different studies.

We fully agree – miRNAs are actually not independent from the sex. Accordingly, in our study we find significantly different miRNAs between males and females. Our intention was to demonstrate that miRNAs have a stronger association with age than with sex. This information was erroneously not mentioned in the heading of the respective results section. We apologize for this error and appreciate that you pointed this inaccuracy. We corrected the headline that now reads “miRNA profiles are stronger associated with the age as compared to the sex” (see also Fig. 2B and Fig. 2C). Since this conclusion is of high importance we now mentioned it also in the abstract: “We observed a larger influence of the age as compared to the sex”.

3. For miRNAs in miRBase database (Version 21.0), why not select the novel version (22.0 or 22.1)? Further, the phenomenon of multiple isomiRs have been widely detected in multiple animal species, especially in diverse human diseases, it is better to show the detailed isomiR landscapes based on the miRNA profiles. As you know, each miRNA locus always yields several dominant isomiRs, and some isomiRs may be involved in diverse sequences or even shifted seeds. I think the systematic analysis based on isomiRs will provide more information for this study.

The underlying technology in our study is microarray analysis. The miRBase version has been changed while our study was running. A change of the content would have also led to a platform bias and batch effects that we try to avoid as much as possible. We decided not to change the experimental system in the running study and to stay with miRBase V21. We want to emphasize that microarrays still have advantages in studies where a large dynamic range of miRNAs is expected. In PAXGene blood samples, 90-95% of all reads are mapping to two or three miRNAs¹⁷. Even if 20 million sequencing reads per sample are mapping to the miRNome, only one million reads remains for all other known 2,500 miRNAs and potentially not yet reported miRNAs. One solution would be a depletion of respective miRNAs as recently reported by the Franke lab¹⁸. Such a depletion in our hands however impacts miRNA expression profiles. We provide more details on this issue in the revised manuscript including an explanation why we used microarray technology.

As you correctly point, isomiRs are a trending and actually very important topic in miRNA research. As isomiRs are also important in our research we recently reported a high-resolution map of human miRNA isoforms¹⁹. As abovementioned the substantial dynamic range in blood however complicates the discovery of low abundant isomiRs. We address these challenges in the revised discussion section of our manuscript. We also want to mention that we are nonetheless interested in aging of isomiRs on a tissue level. We recently, published RNA sequencing and single cell sequencing with the tabula muris consortium^{12,13}. RNA patterns were sequenced for 6 replicates, 17 tissues and 10 time points. For all 1,020 samples we are currently sequencing small RNAs. One of the foremost aims of this study is to understand isomiR aging. We thus appreciate your encouraging comment towards the importance of isomiRs.

Your comment however made us think about whether we evaluated all possible aspects of miRNA analyses that are feasible with our data set. One other trending topic besides isomiRs is so-called arm-shift or arm-switch events. The data we generated are actually tailored to investigate age depend arm-shifts. We observed 40 arm-shift events associated with aging and a general trend to an increased 5' mature expression also associated with increasing age. The results have been added to the revised manuscript and details are provided in the Supplemental Material.

4. More important, miRNA profiles were analyzed based on blood samples from diverse human diseases, including heart disease and lung diseases. However, some of the small RNAs were diverged expressed in diverse tissues. It is quite important to show the temporal expression patterns for these miRNAs.

This is a valid and interesting perspective. We measured whole blood from all diseases to avoid a direct tissue bias or a bias through tissue specific miRNAs scattered to serum (please see also our response to the comment one of reviewer 2). Nonetheless, we of course expect patterns that are diverging because of disease, which affect different organs at different ages. While we tried to find patterns in this annotation matrix (ages x time-points) by SOMs (please kindly see in this context also our reply to comment 10 of reviewer 2) we unfortunately missed to provide details on the divergent expression in the diseases affecting different tissues. In the revised version, we now added the respective information to the section entitled “The association between age and miRNA expression is partially lost in diseases”. Moreover, we provide very detailed information for each miRNA in a new Supplemental Table (Supplemental Table 8). Finally, we also address the challenge indicted by you in the discussion section of the revised manuscript.

5. For figure legends, it is better to show the total description.

We are not exactly sure what you mean by “total description”. We assume that you refer to both, an improved figure legend and to the annotation of the figures, including color codes, axis labels and others. Reviewer one had a similar comment on the figure legends and we kindly ask you to also consider our reply to the reply of comment 5 of reviewer 1. In sum, we improved both, the figure legends and the annotation of figures in the revised manuscript.

6. Finally, the statistical analysis should be performed for the data analysis. Many results (such as Figure 3A, Figure 3b, et al.) should present the detailed p values to show whether it exists significant difference between or among groups.

We are sorry for not having provided accurate p-values and quantification of differences in the original manuscript. Moreover, In the original manuscript we also partially missed to mention the test statistics underlying the p-values. In the revised manuscript we pay attention to have adequate p-values. We also mention the respective test.

For the cases indicated by you, we performed both an analysis of variance and a Wilcoxon Mann-Whitney test as shown in Figure 3A. In both cases the p-value was below the R double precision limit (2.2×10^{-16}). For Fig. 3B we performed the same tests leading to the same results. These and other details have been added to the revised manuscript.

References cited in this letter:

1. Lehallier B, Gate D, Schaum N, et al. Undulating changes in human plasma proteome profiles across the lifespan. *Nat Med* 2019;25:1843-50.
2. Ludwig N, Fehlmann T, Kern F, et al. Machine Learning to Detect Alzheimer's Disease from Circulating Non-coding RNAs. *Genomics Proteomics Bioinformatics* 2019;17:430-40.
3. Kahraman M, Roske A, Laufer T, et al. MicroRNA in diagnosis and therapy monitoring of early-stage triple-negative breast cancer. *Sci Rep* 2018;8:11584.
4. Keller A, Fehlmann T, Ludwig N, et al. Genome-wide MicroRNA Expression Profiles in COPD: Early Predictors for Cancer Development. *Genomics Proteomics Bioinformatics* 2018;16:162-71.
5. Keller A, Backes C, Haas J, et al. Validating Alzheimer's disease micro RNAs using next-generation sequencing. *Alzheimers Dement* 2016;12:565-76.
6. Leidinger P, Brefort T, Backes C, et al. High-throughput qRT-PCR validation of blood microRNAs in non-small cell lung cancer. *Oncotarget* 2016;7:4611-23.
7. Latorre I, Leidinger P, Backes C, et al. A novel whole-blood miRNA signature for a rapid diagnosis of pulmonary tuberculosis. *Eur Respir J* 2015;45:1173-6.
8. Leidinger P, Backes C, Deutscher S, et al. A blood based 12-miRNA signature of Alzheimer disease patients. *Genome Biol* 2013;14:R78.
9. Leidinger P, Keller A, Borries A, et al. High-throughput miRNA profiling of human melanoma blood samples. *BMC Cancer* 2010;10:262.
10. Backes C, Meese E, Keller A. Specific miRNA Disease Biomarkers in Blood, Serum and Plasma: Challenges and Prospects. *Mol Diagn Ther* 2016;20:509-18.
11. Keller A, Meese E. Can circulating miRNAs live up to the promise of being minimal invasive biomarkers in clinical settings? *Wiley Interdiscip Rev RNA* 2016;7:148-56.
12. Tabula Muris C. A single-cell transcriptomic atlas characterizes ageing tissues in the mouse. *Nature* 2020;583:590-5.
13. Schaum N, Lehallier B, Hahn O, et al. Ageing hallmarks exhibit organ-specific temporal signatures. *Nature* 2020;583:596-602.
14. Keller A, Leidinger P, Bauer A, et al. Toward the blood-borne miRNome of human diseases. *Nature methods* 2011;8:841-3.
15. Schwarz EC, Backes C, Knorck A, et al. Deep characterization of blood cell miRNomes by NGS. *Cell Mol Life Sci* 2016;73:3169-81.
16. Cheng L, Sharples RA, Scicluna BJ, Hill AF. Exosomes provide a protective and enriched source of miRNA for biomarker profiling compared to intracellular and cell-free blood. *J Extracell Vesicles* 2014;3.
17. Fehlmann T, Reinheimer S, Geng C, et al. cPAS-based sequencing on the BGISEQ-500 to explore small non-coding RNAs. *Clin Epigenetics* 2016;8:123.
18. Juzenas S, Lindqvist CM, Ito G, et al. Depletion of erythropoietic miR-486-5p and miR-451a improves detectability of rare microRNAs in peripheral blood-derived small RNA sequencing libraries. *NAR Genomics and Bioinformatics* 2020;2.
19. Fehlmann T, Backes C, Alles J, et al. A high-resolution map of the human small non-coding transcriptome. *Bioinformatics* 2018;34:1621-8.

REVIEWERS' COMMENTS

Reviewer #1 (Remarks to the Author):

The authors have addressed all the comments.

Reviewer #2 (Remarks to the Author):

The authors addressed most of the comments and revised their manuscript appropriately.

Reviewer #3 (Remarks to the Author):

The authors made a point-to-point response to the proposed amendments, especially in the amendments to the content of miRNA arm shifts. I agree to accept it.